# A Versatile Approach to Polygonal Object Avoidance in Indoor Environments with Hardware Schemes Using an FPGA-Based Multi-Robot

**DOI:** 10.3390/s23239480

**Published:** 2023-11-28

**Authors:** Mudasar Basha, Munuswamy Siva Kumar, Mangali Chinna Chinnaiah, Siew-Kei Lam, Thambipillai Srikanthan, Narambhatla Janardhan, Dodde Hari Krishna, Sanjay Dubey

**Affiliations:** 1Department of Electronics and Communication Engineering, Koneru Lakshmaiah Education Foundation, Green Fields, Guntur 522502, Andhra Pradesh, India; mudasar.basha@bvrit.ac.in (M.B.); msivakumar@kluniversity.in (M.S.K.); 2Department of Electronics and Communications Engineering, B. V. Raju Institute of Technology, Medak (Dist), Narsapur 502313, Telangana, India; harikrishna.dodde@bvrit.ac.in (D.H.K.); sanjay.dubey@bvrit.ac.in (S.D.); 3School of Computer Science and Engineering, Nanyang Technological University, Singapore 639798, Singapore; siewkei_lam@pmail.ntu.edu.sg (S.-K.L.); astsrikan@ntu.edu.sg (T.S.); 4Department of Mechanical Engineering, Chaitanya Bharati Institute of Technology, Gandipet, Hyderabad 500075, Telangana, India; njanardhan_mech@cbit.ac.in

**Keywords:** multi-robot, collision avoidance, object orientation, behavioral control

## Abstract

Service robots perform versatile functions in indoor environments. This study focuses on obstacle avoidance using flock-type indoor-based multi-robots. Each robot was developed with rendezvous behavior and distributed intelligence to perform obstacle avoidance. The hardware scheme-based obstacle-avoidance algorithm was developed using a bio-inspired flock approach, which was developed with three stages. Initially, the algorithm estimates polygonal obstacles and their orientations. The second stage involves performing avoidance at different orientations of obstacles using a heuristic based Bug2 algorithm. The final stage involves performing a flock rendezvous with distributed approaches and linear movements using a behavioral control mechanism. VLSI architectures were developed for multi-robot obstacle avoidance algorithms and were coded using Verilog HDL. The novel design of this article integrates the multi-robot’s obstacle approaches with behavioral control and hardware scheme-based partial reconfiguration (PR) flow. The experiments were validated using FPGA-based multi-robots.

## 1. Introduction

Unmanned ground mobile robots provide prompt services in certain indoor environments. For the last two decades, individual autonomous mobile robots have provided services that support humans. A recent survey by Globe Newswire [1] indicates that the mobile robot market size is expected to grow from USD 3.36 billion in 2023 to USD 6.94 billion by 2028. These autonomous robots require artificial intelligence to accomplish indoor services with new computational technologies. Current research focuses on two aspects, one related to co-bots (human and machine) and the other related to multi-mobile-robot-based services for various applications in logistics, medicine, and other industries. This paper focuses on the unraveling challenges of multi-mobile robots toward achieving indoor services. Researchers [2] have reviewed various aspects like navigation, sensory fusion, obstacle identification and avoidance, and the challenges posed by autonomous mobile robots.

Sensory information and its fusion play a vital role in evaluating the environment. Sensors like infrared, ultrasonic, PIR, LIDAR, LASAR, and cameras have been used by various researchers [3,4] for the validation of their research. There is a tradeoff between the sensor types, on the one hand, and cost and computational challenges, on the other. These sensors are used for robotic actions such as localization, navigation, and parking. For the last two decades, in the abovementioned studies, robotics have been used with a single type of sensor. The new trend involves the integration of different sensors as per robotics requirements. Ciuffreda et al. [5] discussed the localization of robots by integrating two different sensors: PIR and ultrasonic. The other approach is the integration of similar multiple sensors and framing the sensor fusion to achieve better results, as mentioned by the authors of [2,6]. This sensor fusion allows for the estimation of objects and obstacles. Each obstacle is classified as static or dynamic in the ground environments. Static obstacles are considered as the boundary of the environment [7], and, as per the house/workplace, where objects are essential, including furniture such as chairs and tables in the indoor environment, these are treated as obstacles.

Autonomous robots struggle with the estimation of objects in 2D and 3D. In this respect, the avoidance of obstacles has been addressed by various researchers using Bug, iBug, Bug2, general Voronoi diagram (GVD), and heuristic approaches. Howie Choset et al. [8] have presented various bug algorithms for static-obstacle avoidance. Several recent studies have used dynamic obstacle-avoidance methods [9,10]. Obstacle avoidance is challenging when the multi-robot performs cooperative [11] and distribution [12] tasks. Cooperative robots are dependent on formation and deformation methods. Bai et al. [13] reported on obstacle avoidance with multi-robot formation control, and geometric measures for deformation have been addressed by Aranda et al. [14]. The existing challenges posed by flock groups are performing obstacle avoidance with decentralization and participating in formation once again in the flock group. This challenge of decentralization and obstacle avoidance has been addressed by Choi et al. [15]. The proposed research work addresses centralization to decentralization and vice versa; it performs obstacle avoidance with decentralization using the heuristic-based Bug2 algorithm.

Real-time sensory data were computed continuously to perform the robot’s tasks such as obstacle avoidance, path planning, navigation, formation and deformation behavioral control methods, which required effective computational devices. The new computational technologies are cloud and edge-based systems. CPU (microprocessors), Raspberry Pi, microcontrollers, and FPGA were used in the edge computing part of the study, and cloud computing was performed with GPU and FPGA. The selection of the computing devices was based on the requirements of algorithm dynamics, parallel computation, lower power consumption and cost of the devices. In the last decade, FPGA has been used in robotic applications and obstacle avoidance [16,17,18,19]. A recent survey [20] provides comparative results of FPGA-based robotics, which are effective in real-time applications while consuming less power.

This paper presents the following innovations:Hardware scheme-based algorithms for the identification of obstacle types and their orientation in the indoor environment.The behavioral control mechanism approach has been developed for switching between formation to deformation and vice versa to execute obstacle avoidance.The decentralization of multi-robots using a hardware scheme-based heuristic algorithm to perform obstacle avoidance with respect to the type of the obstacle and its orientation. Partial reconfiguration (PR) flow integration was used to achieve optimized resource utilization on run-time implementation.

This section of the paper elaborates on the literature and related research on multi-robot obstacle avoidance for polygonal objects in an indoor environment. Section 2 describes the hardware scheme-based algorithms and architectures for the proposed approach. The results, which validate the proposed algorithm, are discussed in Section 3. An overall summary of the research contribution is provided in Section 4.

## 2. Hardware-Based Algorithms

This section describes the multi-robot’s hardware-based obstacle-avoidance algorithm for moving around indoor obstacles with different orientations.

### 2.1. Hardware-Based Algorithm for Obstacle Identification and Orientation

Our approach is based on the identification of an obstacle and its orientation. Obstacles that were located in the path of robots required obstacle avoidance. The obstacles are classified based on the position in the view of the robot’s 2D plane. An indoor-environment object mostly manifests visually as a plane surface contour, as shown in Figure 1a. Some objects, such as chairs and sofas, have integrated depth and plane contours, as shown in Figure 1b.

Algorithm 1 presents details of the identification of the obstacles in the environment and their orientation. Each robot interfaced with six sensors; S_FT_ and S_FM_ appear on the front side of the robot. All the sensors will sense the environment with the distances as mentioned in Line 1. To avoid a collision, the maximum (max) and minimum (min) distances are assigned as 2 and 1 m, respectively. In the 2D view, the robots’ front sensors estimate which path is free and which contains obstacles (Line 3). When an obstacle is in the way, the robot evaluates what type of object it is, such as cupboards and tables, as shown in Figure 1a (Line 5). When the distances detected by both front sensors are differentiated, S_FT_ will estimate the depth of the object and S_FM_ will establish the plane of the object (Line 6). This sensory fusion confirms whether there are chairs and sofas in the environment as mentioned in Figure 1b. When sensory distances are determined, S_FT_ maximum and S_FM_ minimum define such objects as small stools (Line 7). In this environment, the objects are positioned with different angular orientations (Line 9–13). The sensors estimate the object orientation with respect to the present plane, and object identification is performed in a similar way to that mentioned in Lines 2–7.
**Algorithm 1:** Pseudo code for identification of obstacle and orientation 
1.    Initialize sensory distance and reference distances
2.    Case (Obstacle)
3.    State_1: if ((S_FT_ && S_FM_) > dmax_ϑ_0_ ^0^)? forward: State_2;
4.    State_2: if ((S_FT_ && S_FM_) ≤ dmax_ϑ_0_ ^0^)? State_3: State_5;
5.    State_3: if ((S_FT_ == S_FM_) ≥ (dmin_ϑ_0_ ^0^))? Alg_2@ Case_1: State_4;
6.    State_4: if ((S_FT_ ≥ dmin_ϑ_0_ ^0^) && (S_FM_ < dmin_ϑ_0_ ^0^))? Alg_2@ Case_1: Orientation; 
7.    State_5: if ((S_FT_ ≥ dmax _ϑ_0_ ^0^) && (S_FM_ ≥ dmin_ϑ_0_ ^0^))? Alg_2@ Case_1: Orientation; 
8.    end case
9.     Case (Orientation): 
10.    State_11: if ({S_FT_, S_FM_} @ ϑ ± _15_ ^0^, ±_30_ ^0^, ±_45_ ^0^, ±_60_ ^0^, ±_75_ ^0^) ≥ dmin)? Alg_2@ Case_2: State_12;
11.     State_12: if ((S_FT_ ≥ dmin_ϑ ± _15_ ^0^, ±_30_ ^0^, ±_45_ ^0^, ±_60_ ^0^, ±_75_ ^0^) && (S_FM_ < dmin_ϑ ± _15_ ^0^, ±_30_ ^0^, ±_45_ ^0^, ±_60_ ^0^, ±_75_ ^0^))? Alg_2@ Case_2: State_13; 
12.    State_13: if ((S_FT_ ≥ dmax_ϑ ± _15_ ^0^, ±_30_ ^0^, ±_45_ ^0^, ±_60_ ^0^, ±_75_ ^0^) && (S_FM_ ≥ dmin_ϑ ± _15_ ^0^, ±_30_ ^0^, ±_45_ ^0^, ±_60_ ^0^, ±_75_ ^0^)), Alg_2@ Case_2;
13.    end case

#### 2.1.1. Hardware-Based Algorithm for Obstacle Avoidance of Indoor Polygonal Objects 

Figure 2a–d and Algorithm 2 represent the obstacle avoidance of the multi-robot for various indoor environment obstacles with different orientations. Algorithm 2 collects the information from Algorithm 1 and confirms an obstacle and its orientation. In the robot’s view, there is an obstacle with a 2D plane in the right orientation, as presented in case_1, and different orientation object avoidance, as presented in case_2. In case_1, obstacle avoidance is performed in the Bug2 algorithm with odometer techniques (Lines 2–11). Both front sensors’ distance is equal to the minimum distance in the line of sight (dmin_ϑ_0_ ^0^) (Line 3), and it is evaluated based on the Euclidean distance method. Using the rendezvous approach, multi-robots, while performing obstacle avoidance, execute as left flock (F_L_), which is positioned on the left side, whereas right flock (F_R_) is positioned on the right side among the group. The flock group performs a ϑ_90_^0^ turn with respect to their side from dmin_ϑ_0_^0^ values and applying a wall-following approach, which continues until it reaches the edge of the object (Line 4). In parallel, the internal soft odometer evaluates the distance traveled from the turn position to the edge (Line 4). The flock team, F_L_ and F_R_, takes a ϑ_90_^0^ turn right and left (Line 5). The proposed research work addresses obstacle avoidance for polygonal objects such as rectangles, squares (four sides), triangles (equilateral), and these polygonal objects are classified by the flock group using right (S_R_) or left (S_L_) sensory distance with respect to the object plane (Line 6). The flock group evaluates the distance and follows the object with a parallel (rectangle/square) or perpendicular (triangle) movement (Lines 7–10). Based on the edge of the object, the flock team takes a ϑ_90_^0^ turn towards the object side and forward to crucial part of the Bug2 point. In this heuristic approach, odometer distance evaluated in the initial stage will be decremented until equal to the null value (Line 11). This technique is one of the novel approaches for a multi-robot flock performing obstacle avoidance. Moreover, case_2 has similar lines to case_1 (Line 13). In this case, the robots move in parallel to the object until they reach the edge, while the odometer records the distance and evaluates the angle from the initial position (dmin_ϑ_0_ ^0^) to the edge of the object (ϑ_x±_ ^0^) (Line 14). The flock group performs the turn with respect to the invert value of ϑ_x±_ ^0^ (Line 15). The next stage of obstacle avoidance is performed in Lines 6–11.
**Algorithm 2:** Pseudo code for obstacle avoidance 
1.    Initialize obstacle identification and orientation
2.    Case_1 (Obstacle avoidance)
3.    State_1: if ((S_FT_ == S_FM_) ≥ (dmin_ϑ_0_ ^0^))? State_2: Case_2.
4.    State_2: turn ϑ_90_ ^0^ F_L_ -> L & F_R_ -> R; Wall follow (odometer++),
5.            turn @edge ϑ_90_ ^0^, F_L_ -> R & F_R_ -> L.
6.    State_3: if ((F_L_ (S_R_)) && (F_R_ (S_L_)) = dmin_ϑ_0_ ^0^)
7.          Wall follow in Parallel to object, Take turn @edge ϑ_90_ ^0^ F_L_ -> R & F_R_ -> L, 
8.          else
9.          Wall follow in Perpendicular to object, Take turn @edge ϑ_90_ ^0^ F_L_ -> R & F_R_ -> L.
10.        end 
11.      Forward (odometer - -), Take turn ϑ_90_ ^0^ F_L_ -> L & F_R_ -> R, end case.
12.     Case_2 (Obstacle avoidance _ Orientation): 
13.    State_11: if ((F_L_ (S_FT_)) && (F_R_ (S_FT_)) ≠ dmin_ϑ_0_ ^0^)? State_12: Case_1. 
14.    State_12: turn w.r.t to object orient ϑ_x±_ ^0^, F_L_ -> L & F_R_ -> R; Wall follow (odometer++).
15.           turn @edge ϑ_x±_ ^0^, F_L_ -> R & F_R_ -> L.
16.   State_13: Repeat; State_3.
17.    end case

#### 2.1.2. Formation and Deformation of Multi-Robot in Indoor Environment 

Figure 3a,b presents the multi-robot behavioral control mechanism for a formation of multi-robots with a rendezvous approach and deformation into linear form at dynamic situations in the environment. When robots navigate through the indoor environment past objects at different orientations, they are able to transform from formation to deformation flock approaches. F_xy_ denotes the flock (F), the side (x) and the role (y) of the group; thus, in the four-robot group, two are on the left side and the other two are on right side. Preferably, front robots are considered as leaders, and those behind are enrolled as followers. Figure 3b presents the network topology between the multi-robots based on the static and dynamic conditions of the environment. Upon entry into the environment, leader robots communicate with each other and maintain a flock group by sharing information about the environment. When an obstacle like a sofa appears in front of the leader robots at node A, the flock group makes the decision to traverse in their respective directions and, like the leaders, communicates with other robots. Meanwhile, leaders exchange details of their respective angles and odometer information. This information matches their odometer and angles based on the previously reached group, which waits for the next flock group at the end point. In this regard, after obstacle avoidance at node A, it reforms back in its original location as the rendezvous formation group moves forward. In certain conditions, such as node B, multi-robots cannot perform the operation of either the flock group or the rendezvous group. Multi-robots transform from formation to deformation or decoupling, i.e., robots reframed with a bio-inspired system including ant movements, into a line approach. In this process, the robots F_LL_, F_RL_, F_LF_ and F_RF_ move in an odd-and-even-number sequence. Upon completion of this sequence, the sequence transforms into formation using a first-in first-out (FIFO) approach. The flock of robots at node D is reached through node C, which contains a triangle-shaped object with a different orientation that incorporates prior algorithms and behavioral control. It is similar in terms of formation after the obstacle avoidance of the multi-robot.

As per our proposed approach, a finite-state machine for the algorithms is presented. The proposed model consists of four main states. Obstacle Detection: It identifies the object with respect to the sensor(s) data received. Obstacle Orientation: The main function is to identify obstacle orientation for flat and polygonal shapes. Obstacle Avoidance: To perform the action, avoiding the obstacle is the functionality of this state by performing robot maneuvers and interchange. Obstacle Avoidance and Orientation: To correct its orientation (dmin_ϑ = {15°, …, 75°}) with respect to the identified object so that the flock of robots can navigate through the environment. The left half of the FSM shows obstacle detection and orientation, and the right half of the FSM shows obstacle avoidance and avoidance orientation, as shown in Figure 3c. A detailed explanation is provided below.

The framework for the flock of mobile robots is the reset condition in which the control unit fetches the instruction through the “Obstacle Detection and Orientation” state. According to this case, the robots first set up their sensors and system while they are in the “Initialization” condition. The next step is “Obstacle Avoidance (Case_1)”, which is meant to help recognize and avoid things that have flat surfaces. In this scenario, the robots transition through several states, including “State_1”, where they detect the presence of an object at node point A, and “State_3”, where they perform actions according to the proximity of the object. When the robots discover an object that is irregular and requires more advanced navigation, they have the option to switch to “Obstacle Avoidance & Orientation (Case_2) at node B”. Here, they pass through states such as “State_11”, which permits them to modify their orientation in relation to the object they have identified, and “State_12”, which enables them to avoid obstacles while preserving their orientation.

The obstacle detection and navigation system depicted in the state transition graphic (right half) is centered around two primary scenarios: “Obstacle Avoidance (Case_1)” and “Obstacle Avoidance and Orientation (Case_2)”. This system starts at the “Initialization” state upon reset of the condition to the “Obstacle Avoidance (Case_1)”, where the sensors and actuators are initialized. With SFT and SFM sensor information is greater than the minimum distance, the transition to “State_1” happens, signaling that an object has been identified at corresponding orientation degree. The system determines whether the distance of the detected object falls under a predetermined range in “State_1”, at node B to node point C the algorithm switches to “State_2” to enable maneuvers like turning in an angle of 0° to 90° with wall following as FL -> L & FR -> R. In order to handle increasingly complicated items or circumstances, the system falls back to “State_11” if the condition is not satisfied. When the barrier has been overcome, the looping state “State_3” can return to “Obstacle Avoidance (Case_1)” or return to “State_3” where wall following and turns are carried out according to the object’s properties as FL -> R & FR -> L.

To correct its orientation with respect to the identified object, the system switches from “State_11” to “State_12” in “Obstacle Avoidance and Orientation (Case_2)”. Turning and wall following may be necessary for this at node C to node D. The system goes back to “State_13” and continues the obstacle avoidance and orienting procedure with the help of internal odometer incremented states to store the odometer value in the FIFO. Once the data are retrieved from the FIFO, the odometer value is decremented to deal with impediments effectively, and the system can then carry on with Obstacle Avoidance in “State_3” of “Obstacle Avoidance (Case_1)” by returning to case_1 of obstacle avoidance and checking for necessary action to be triggered to control the actuators.

The transitions and states of this model make it easier for the robot to maneuver around objects of varying sizes and orientations. It is a flexible method for obstacle recognition and avoidance in a dynamic environment because it can easily transition between situations and adjust to flat and irregular barriers. Hence, a flock of robots can navigate their surroundings efficiently and handle barriers that are both flat and uneven by looping through these states as needed.

### 2.2. Hardware Schemes

The proposed method features hardware-based pipeline architectures for the decentralized and distribution-type multi-mobile robot obstacle avoidance, as shown in Figure 4. This architecture is deployed in the programmable logic (PL) of the FPGA. The architecture consists of two aspects, inter and intra module, which are integrated into each mobile robot. Environments with and without obstacles were analyzed based on sensory information.

Obstacles are classified as static or dynamic; the proposed contribution focuses on the estimation of a polygonal static obstacle and its orientation. The inter module presents the actions within an individual robot, such as interfacing ultrasonic sensors and triggering at every 1 ms, and pulse-width modulation (PWM) echo signal converted into the 32 bit distance using PWDC_sensor fusion. This module consists of six ultrasonic sensors, and each sensor FIFO size is 32 × 64. The internal architecture frames the sensor fusion using the FIFO data, the same as is transmitted for the other modules in the system. The ESP8266 Wi-Fi module has been integrated through a UART module with a baud rate of 9600, which is utilized while performing the behavioral control mechanism in avoiding collisions and positioning among the flock group. The servo motors play a more significant role in the estimation of obstacle orientation: ultrasonic sensors are positioned on top of the servo motors and as per control unit instruction, it rotates with a step size of 15° toward the right and left up to ±90°. The hardware scheme algorithms employ a pipeline architecture that consists of a control unit which drives the inter and intra modules.

The intra module of the architecture is embedded with obstacle identification and orientation modules, obstacle avoidance and the behavioral control mechanism module, along with the partial reconfiguration control module. As per hardware scheme-based Algorithm 1, an equivalent architecture is the obstacle identification and orientation module. Based on the sensor fusion, this module estimates the type of polygonal obstacle such as rectangular and triangular. The estimation of the obstacle orientation is challenging, as the orientations are considered with respect to the robot in the environment. These 8 bits comprise primary data that extend their operations to accomplish their purpose in the architecture. Obstacle avoidance is central to this research; it was developed based on the hardware scheme for Algorithm 2. This architecture addresses obstacle avoidance for static objects with and without orientation as per the environment. Dynamic objects include humans and other robots. A human walking at a normal speed of 1.44 m/sec and the other robots’ speed are known information; with this information, delay units are developed until they reach the robots. The behavioral control mechanism is a state-of-the-art architecture, which is event driven based on switching from formation to deformation and vice versa. This module depends on sensory information, event-driven conditions with respect to the obstacle type and avoidance, and robot localization among the flock group. The flock group will maintain a minimum of 50 cm between team members who are positioned in front or behind and to the left or right.

Deformation is the switching module, which links centralization to distribution to perform obstacle avoidance. After accomplishing obstacle avoidance, the module switches from distribution to centralization, which is embedded in the formation module. The event-driven conditions are replicated by the partial reconfiguration module, which drives the respective module based on the event or situation. It is one of the novel approaches to implementing hardware schemes for multi-mobile robot obstacle avoidance. This approach controls some modules as active, while others are in sleep mode; this decreases power consumption due to the lack of computation by modules in sleep mode. The execution unit is driven based on the modules’ activities and respective motor actions are mapped and operated. In this mobile robot, two motors are positioned on its left and right, and respective actions are performed as per the instructions of the execution unit.

#### 2.2.1. Hardware Schemes of Obstacle Identification and Orientation

Figure 5 illustrates the internal architecture of the polygonal type of obstacle identification and its orientation. This module consists of two processing elements (PEs): PE_1 performs obstacle identification and shape, PE_2 provides evaluation of obstacle orientation. The sensory fusion data are collected from the FIFO and directed towards CORDIC modules and the direction-based sensor distance module. The direction module switches the sensory information using a shuffle network approach to assign the distances to S_FT_ and S_FM_ based on the current position versus objects in the environment. The Xilinx CORDIC IP core was utilized in this module to generate angles of sensory distance and was assigned based on the requirements of S_FT__ϑ_x_^0^ and S_FM__ϑ_x_^0^. The predefined distance is calculated based on the Euclidean distance approach and is prestored in FIFO array 2 × 32 × 10; dmin and dmax are two reference distances with respect to 10 angles such as ϑ ± _15_^0^, ±_30_^0^, ±_45_^0^, ±_60_^0^ and ±_75_^0^ with 32-bit width. Initially, the hardware confines either obstacle presence or obstacle absence. In the absence of an object, it instructs the execution unit to take forward action. When an object is present, it switches to PE_1; when the conditions are satisfied in PE_1, it drives to the obstacle avoidance module with respect to various conditions; it is presented using a digital encoder in this design. The PE_2 presented in Figure 5 represents obstacle orientation. The obstacles are oriented towards an obtuse or acute angle from the perception position. The set of digital rulers provides a solution for the identification of object orientation. PE_2 consists of the 10 array modules for evaluating the angles ϑ ± _15_ ^0^, ±_30_ ^0^, ±_45_ ^0^, ±_60_ ^0^ and ±_75_ ^0^. It uses the matching approach with respect to the reference dmin_ϑ_x_ ^0^, dmax_ϑ_x_ ^0^ and real-time distances S_FT__ϑ_x_^0^, S_FM__ϑ_x_^0^. Once the orientation of the object is confirmed, it progresses to the next stage of the system. The digital encoder designed with 30 (3 i/p each module @10 folds) inputs and 5 lines of output is used to share information in the same way as the obstacle avoidance module.

#### 2.2.2. Hardware Schemes of Obstacle Avoidance for Distributed Multi-Robots

The obstacle avoidance of the proposed research work was based on Bug2 lines with integration of Euclidian distance measurement and real-time digital computation using CORDIC [21]. Figure 6 presents the internal architecture of PE_11; after obstacle identification, the robot takes angle ϑ_90_^0^, and it is designed by using the counters. The counter integral part of the execution up to ϑ_90_^0^ and proportionality verifies the real-time angle movement of the robot using the CORDIC modules versus the reference value (ϑ_90_^0^ training data defined as event). These angle actions are performed at both edges of the polygonal object as per the algorithm. While performing a forward action from a ϑ_90_^0^ turn to the next edge, the hardware increments the counter as odometer ++, which registers this information. PE_12 is similar to PE_22 without the angle orientation of the objects as mentioned in Figure 7. In the next stage, PE_12 follows two actions, one of which follows the boundary of the obstacle and recognizes with sensor fusion if the object is parallel (rectangular polygon object) to the robot’s position. Concerning either surface with inclination, its distances vary for every forward step by the robot; then, Euclidean distance and computation are performed to continue to the next edge of the object, and next, a ϑ_90_^0^ turn is performed. From this, the odometer’s previous value is a decrement for every step size until the odometer -- reaches a null value. PE_21 operates in line with PE_11 based on the orientation of the object, and PE_22 performs as PE_12 does in terms of object orientation. The CORDIC module is used a number of times to provide the distance with respect to angles. The robots from this position switch back to formation among the robots using the behavioral control mechanism in similar a way to that described by Divya et al. [22]. The robots are positioned in a first-come, first-served approach: the front-line robots move forward and create space for follower robots in the group as per the behavioral approach. The total design consists of four processing elements as mentioned in Figure 8: PE_11 and PE_12 determine the obstacle avoidance for normal cases such as ϑ_0_ ^0^, PE_21 and PE_22 work towards orientation-based obstacle avoidance.

## 3. Results

The results presented in this article take the form of a resource utilization with synthesis report, and a Xilinx-based FPGA was used for the experimental setup and experimental validation. Xilinx Vivado 2017.3 was preferred for the complete cycle of the implementation. Verilog HDL was used for scripting the hardware scheme equivalent code and was simulated, synthesized, and implemented on a Zed board FPGA.

### 3.1. Resource Utilization

The hardware schemes for polygonal object avoidance were developed with the equivalent code for multi-robots. The individual modules with HDL were developed and integrated with AXI lite for system integration. The obstacle identification modules were framed with combinational circuit designs, obstacle avoidance and communication between robots dependent on a 100 MHz clock frequency, which was synchronized using the AXI lite type system integration. The novel approach was developed in this article using partial reconfiguration for decreasing the power consumption. This state-of-the-art approach is the first of its kind for multi-mobile robot obstacle avoidance. This reconfiguration was performed on a run time with respect to an event-driven situation, and the control unit plays a vital role in the execution of the dynamic partial reconfiguration module.

Table 1 presents the device/resource utilization of the proposed approach in this article. The Zynq XC7Z020 (Xilinx, San Jose, CA, USA) device was used for deployment of the Verilog HDL. Its overall resource of FPGA is 53.2 K look-up tables (LUT), block RAM 140 (4.9 Mb), and DSP slice with 220. Figure 9a presents a resource utilization summary of general and PR flows with respect to an event or situation. The PR plays an important role in the behavioral control mechanism as the multi-robots switch from formation to deformation, and vice versa.

The general flow presents the resource utilization of LUT, BRAM, and DSP slice as 67%, 84% and 65%, respectively. We observed that BRAM occupied a higher level, affecting the performance in relation to run time. The hardware resources are limited in such an evaluation board as the Zed board; if a switch to a high-end board is desired, there is a trade-off in terms of cost versus resource availability. In this regard, we considered utilizing partial reconfiguration (PR) flow, and it has since been approved by researchers [23,24,25]. The PR flow provided interesting results, which were observed by using the Xilinx Integrated Logic Analyzer (ILA) while estimating the performance of the device. The PR values in different events or situations with LUT, BRAM and DSP slice were as follows: obstacle identification 38%, 52% and 46%; obstacle avoidance 48%, 60% and 56%; behavioral control mechanism 44%, 52% and 48%. The PR impacts static power consumption as mentioned in Figure 9b; general flow uses 2.4 watts, whereas PR flow uses an average of 1.8 watts, individual module power consumptions are obstacle identification 1.65 watts, obstacle avoidance 1.95 watts and behavioral control 1.8 watts. These data were captured using the Xilinx Power Estimator (XPE) and Vivado tool.

### 3.2. Experimental Results

The multi-robots were designed using a CAD model that is suitable for indoor environment services. In this study, four non-holonomic robots were used for the experimental validation of multi-robot obstacle avoidance using the formation and deformation of the robots’ approach. The robots’ mechanical hardware consisted of three levels of circular plates, and the ground level in the robot was established with two 24 V/7 A batteries. The intermediate level comprised computational devices and sensors. The ultrasonic sensors were positioned on the four sides of the robot, and two other sensors were positioned between the front and the right and left sides.

Two stepper motors were interfaced to each robot for both sides with wheels as illustrated in Figure 10a,b. Servo motors were positioned under the sensors, which are useful for the estimation of object orientation. The top layer was used for service applications, such as carrying food and other materials.

The FPGAs were powered by batteries using a 7805-voltage regulator. Stepper motors were powered with two 24 V/7-amp batteries (coupled in series to generate 24 V). The flock between robots was performed based on their dimensions, as shown in Figure 10b. The robots were localized to the boundary and maintained distances X_d_ and Z_d_. The other robots maintained the distance between the boundary robots as a delta. The distance between the flocks was maintained at 1 to 1.5 × the robot diameter in all directions, and it made sense to perform kinematic movements of the mobile robots. Every angular movement and distance were computed using an algorithm based on CORDIC.

The proposed research has two sets of experimental results: The initial method presents obstacle identification, obstacle avoidance and behavioral control, which allows the robot flock group to transition from formation to deformation and reform into a linear formation flock group. The other method involves identification of the obstacle with its orientation, which performs respective obstacle avoidance. At the same time, behavioral control transitions from linear formation mode to deformation with decentralization obstacle avoidance and finally reaches the flock group.

Experimental results of multi-robot obstacle identification and avoidance

Figure 11a–f illustrates experimental validation for the obstacle identification and avoidance of the multi-robots. According to the algorithm, the foremost action among the robots is the identification of their role based on the location in the flock group using the behavioral control module. The front-line robots lead their respective side and the back-line robots act as followers in the flock group. Figure 11a presents the leader robots analyzing the object type and its orientation, which is communicated to their team members as shown in Figure 3b. The robots transition from formation to deformation and perform obstacle avoidance with a distributed approach, as illustrated in Figure 11b–e. Obstacle avoidance was performed with Algorithm 2 and using the architectures shown in Figure 6 and Figure 7. This obstacle avoidance was performed using the Bug2 approach; our proposed integration of the soft odometer was designed to achieve successful obstacle avoidance. Figure 11f presents an environment that is blocked on both sides with a narrow free space between the objects. This is another state-of-the-art aspect of the multi-robot behavioral control switch from deformation to linear formation (robots move back-to-back based on their respective role or approach).

In Figure 11f, the four robots fulfilled their role until there was a maintained delay to avoid collisions between them. Figure 12a–h provides snapshots of the experimental results of the multi-robots moving from linear formation to deformation, before reaching the flock group rendezvous and returning to formation.

Experimental results of multi-robot identification of oriented objects and avoidance.

In this process, as shown in Figure 3b at node c, the robots identify the object with angular orientation; they perform obstacle avoidance using the PE_21 and PE_22 architectures mentioned in Figure 6, Figure 7 and Figure 8. The robots moving on the right side of the object move parallel to the wall as wall followers, and on the other side, the robots take a perpendicular approach. Once the robots reach the end of obstacle avoidance (after odometer −−), the leader robots move forward and create space for the followers to satisfy the flocks’ roles by restoring the formation among the robots. This dynamic and varied object-based obstacle avoidance is successfully performed using FPGA-based robots and PR flow. Experimental validation provided in the public link: https://www.youtube.com/watch?v=KmKvLk-DJOw (accessed on 11 October 2023).

Table 2 presents a comparison of the various methods employed in the field of multi-robot obstacle avoidance. The parameters are considered based on how essential they are for obstacle avoidance. Various sensory approaches have been addressed by the researchers; the proposed approach took ultrasonic sensors into account to avoid resource issues in FPGA implementation. The vision-based obstacle avoidance methods require higher FPGA, which is more costly and creates computational issues such as pose estimation. Optimized algorithms are required for vision-based obstacle avoidance in the future. Based on the comparison to other methods, for the proposed method, we selected FPGA and partial reconfiguration flow for better results, as presented in Table 1. Multi-robot behavioral control reforming approaches also feature in this table as formation-to-deformation (or vice versa)-based event conditions.

The experiments conducted with different perspectives in various environmental scenarios to calculate the error rate of the proposed algorithm are tabulated in Table 3. Obstacle identification in the angular orientation is the hot core of the proposed approach, and the error rate is reduced to 3.8%. The error rate is much improved from node C to node D when the flock of robots forms a linear shape as the formation approach changes to a deformation approach and cross the obstacle in an odd-and-even-number sequence method with 3.2%. At node D, the error rate is possibly reduced when the flock of robots takes a deformation-to-formation approach. Figure 13 shows the statistical analysis of the proposed algorithm, which enhances the efficiency of our algorithm in estimating the error rate compared to previous approaches. In Table 2, X stands unavailable and √ represent available of sensor fusion.

## 4. Conclusions

In this study, multi-robot obstacle avoidance was performed using hardware schemes and a behavioral control mechanism. Obstacle avoidance depended on the identification of obstacles and their orientation. Multi-robot flocks depended on their behavioral control mechanism, which allowed them to perform avoidance without colliding with other robots, using formation and deformation along with decentralization methods to achieve better results in multi-robotic fields. The integration of these methods was performed with a reconfigurable computing device, FPGA, using partial reconfiguration flow. The multi-robots performed obstacle avoidance with respect to any situation or event such as flock formation, linear formation, and deformation. This article attempted to provide experimental validation using FPGA-based robots. Each robot was competent in executing the general FPGA flow and PR flow. The PR flow provided better results in the format of LUT, BRAM, and DSP slice: obstacle identification of 38%, 52% and 46%; obstacle avoidance of 48%, 60% and 56%; and behavioral control mechanism of 44%, 52% and 48%, respectively. This represents state-of-the-art research by the integration of hardware schemes with PR flow for behavioral-control-based obstacle avoidance. The PR impacts the static power consumption in general flow by 2.4 watts, whereas PR flow uses an average of 1.8 watts.

## Figures and Tables

**Figure 1 sensors-23-09480-f001:**
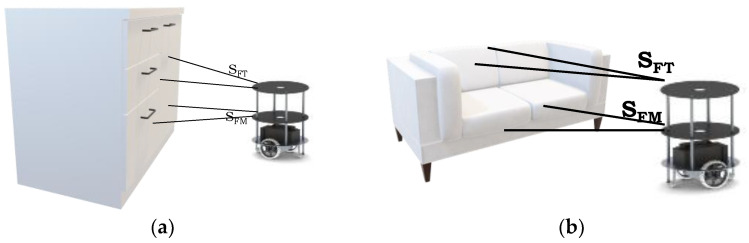
(**a**,**b**) Indoor-environment-based polygonal object identification and orientation.

**Figure 2 sensors-23-09480-f002:**
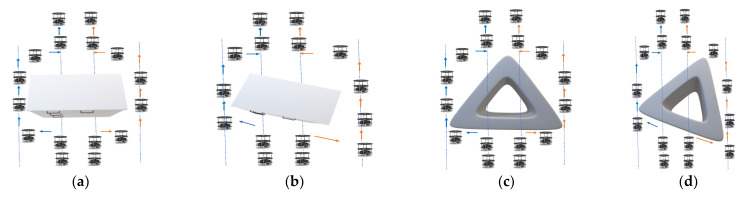
(**a**–**d**) Multi-robot obstacle avoidance for polygonal objects with different orientation.

**Figure 3 sensors-23-09480-f003:**
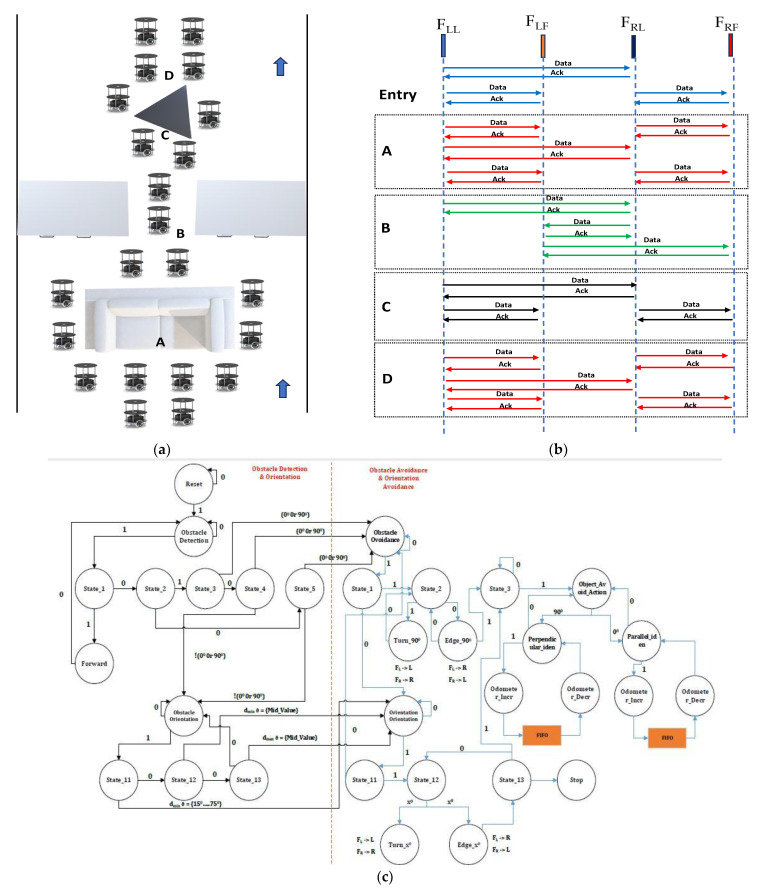
(**a**,**b**) Multi-robot behavioral control mechanism for formation and deformation. (**c**) Finite state machine of proposed algorithm.

**Figure 4 sensors-23-09480-f004:**
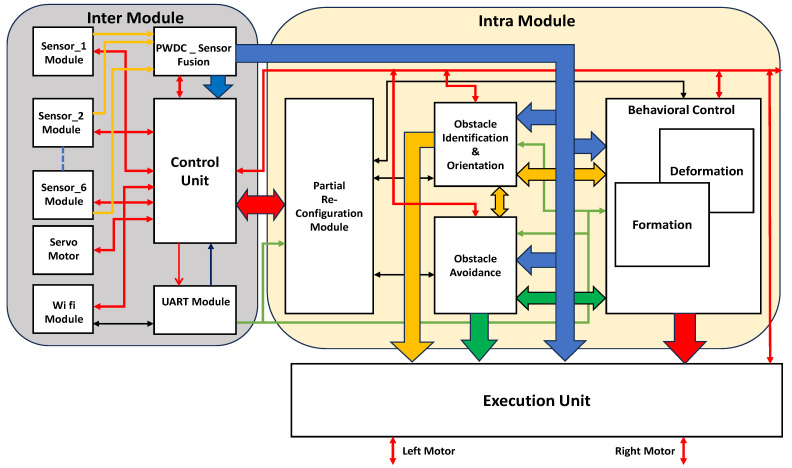
Hardware scheme for distribution multi-mobile robot obstacle avoidance.

**Figure 5 sensors-23-09480-f005:**
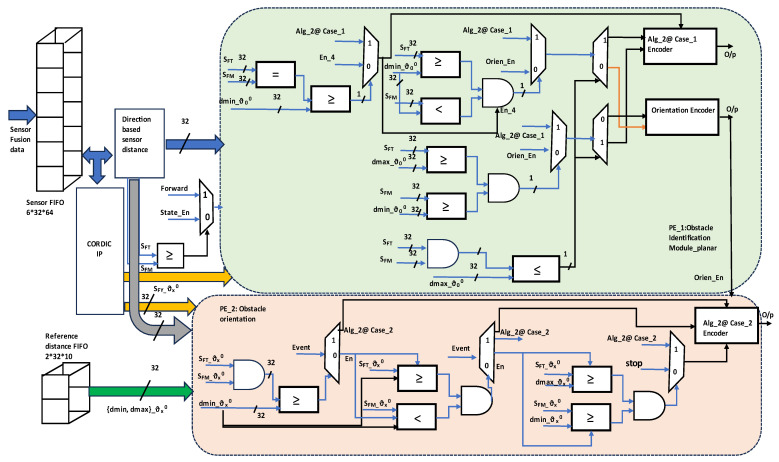
Architecture of obstacle identification and orientation in an indoor environment.

**Figure 6 sensors-23-09480-f006:**
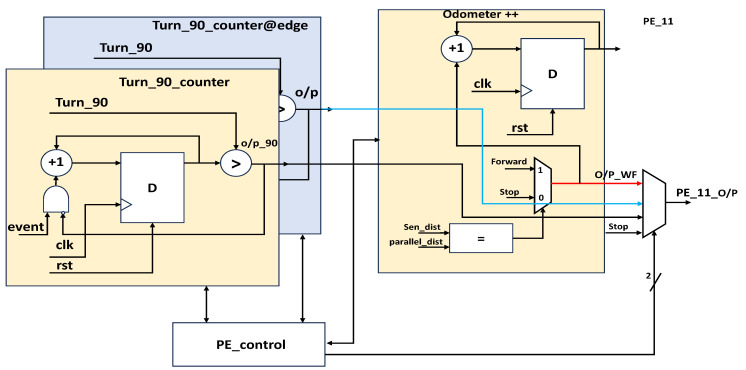
PE_11 for obstacle avoidance at initial stages and intermediate stages.

**Figure 7 sensors-23-09480-f007:**
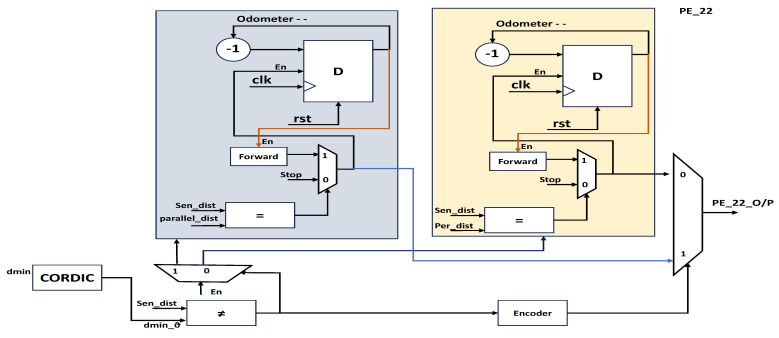
PE_22 final stage for angle orientation-based obstacle avoidance.

**Figure 8 sensors-23-09480-f008:**
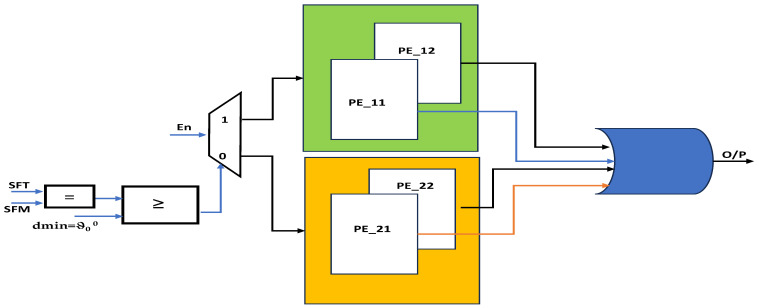
Hardware scheme for obstacle avoidance using PEs.

**Figure 9 sensors-23-09480-f009:**
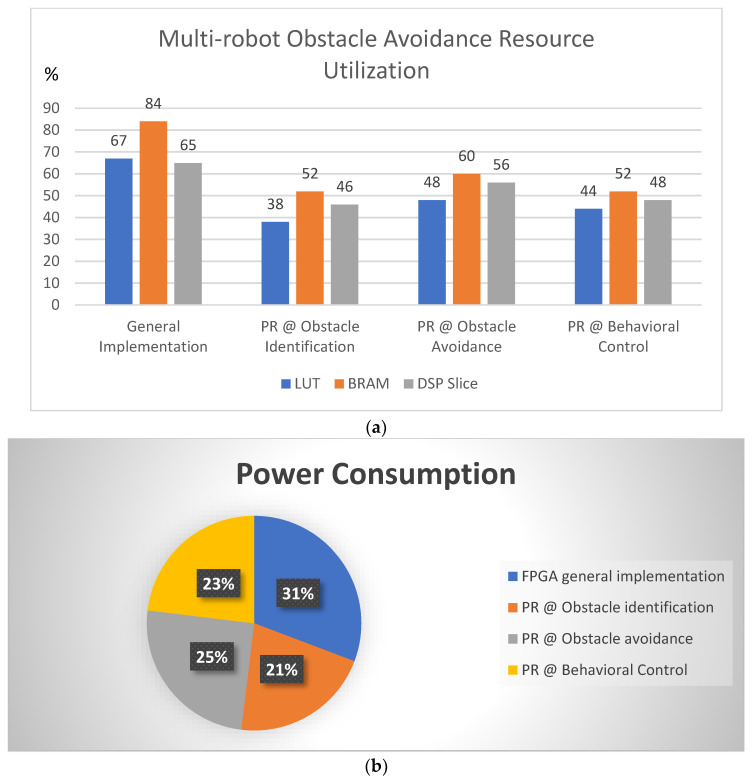
(**a**) Multi-robot obstacle-avoidance resource utilization in general and PR flow. (**b**) Power consumption for multi robot obstacle-avoidance in general and PR flow.

**Figure 10 sensors-23-09480-f010:**
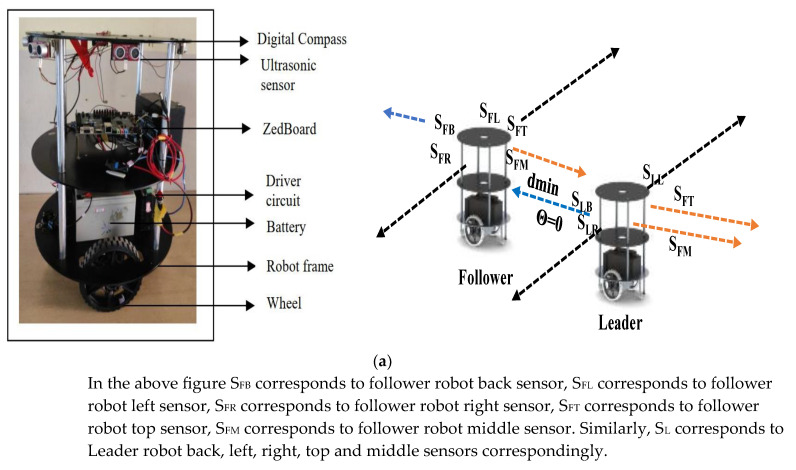
(**a**) Mobile robot experimental setup. (**b**) Multi robot flock.

**Figure 11 sensors-23-09480-f011:**
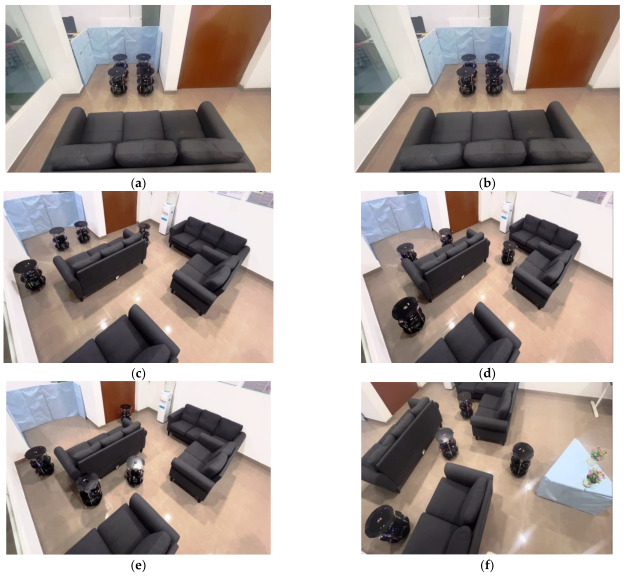
(**a**–**f**) Experimental results of obstacle identification and avoidance by multi-robots.

**Figure 12 sensors-23-09480-f012:**
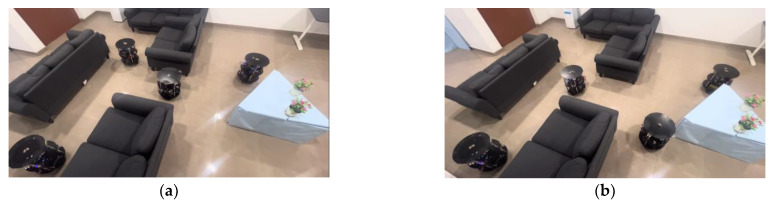
(**a**–**h**) Experimental results of obstacle identification with orientation and avoidance by multi-robots.

**Figure 13 sensors-23-09480-f013:**
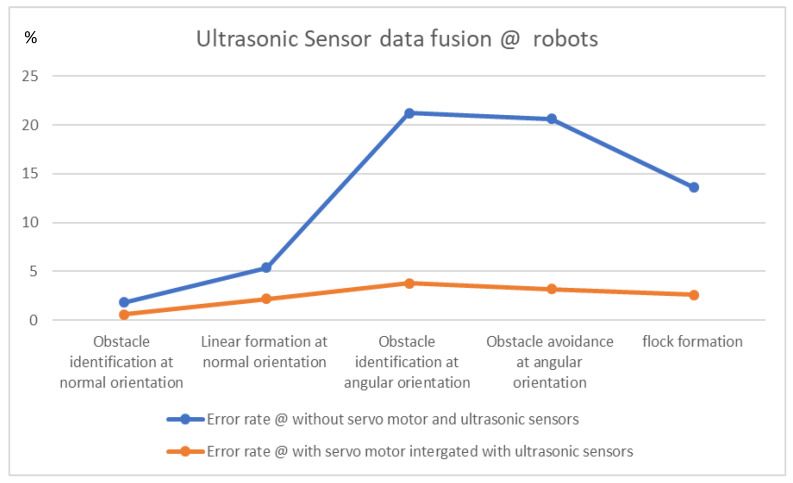
Ultrasonic sensor fusion data error rate of proposed algorithm.

**Table 1 sensors-23-09480-t001:** Resource utilization for multi-robot obstacle avoidance.

Module	LUT	BRAM	DSP Slice
Obstacle Identification and Orientation	3724	18	8
Obstacle Avoidance	8416	28	30
Interfacing Modules (sensors, motors, communication, Xilinx IP cores)	6852	24	36
Control Unit and PWDC Sensor Fusion	4468	20	42
Partial Reconfiguration Module	5586	12	14
Behavioral Control Module	6628	16	12
Total	35,674	134	142

**Table 2 sensors-23-09480-t002:** Comparison of multi-robot obstacle avoidance with relevant research methods.

Reference Papers	Sensory Approach		Algorithm	Hardware	Pros	Cons
	Method	Fusion				
[26]	RGB-D camera	X	Multi-attribute decision making	CPU	Reinforcement learning (RL)	Limited to simulation
[27]	Virtual force	X	Hybrid force/ position	CPU	Fuzzy adaptive controller	Limited to simulation
[28]	LIDAR	X	Multi-robot collision avoidance	CPU	Leader–follower formation control	Higher power consumption
[29]	-	X	Nonlinear model predictive control	CPU	Dynamic obstacle avoidance	Limited to simulation
[30]	_	X	Dynamic obstacle avoidance of differential-drive wheeled mobile robot	CPU	Skidding and slipping analysis in obstacle avoidance	Limited to simulation
[11]	Ultrasonic sensor	X	Centralized obstacle avoidance	FPGA	Hardware schemes for centralized multi-mobile robot’s obstacle avoidance	Partial reconfiguration not part of the hardware design
Proposed	Ultrasonic sensor	√	Centralization at formation and distribution at deformation method for obstacle avoidance	FPGA	Partial reconfiguration-based hardware schemes are a novel approach	Velocity-based obstacle avoidance will be addressed in future

**Table 3 sensors-23-09480-t003:** Ultrasonic sensor data fusion error rate at various scenarios.

Environment Scenario	Ultrasonic Sensor Data Fusion	Capture Sensory Data Fusion @ Positive Rate	Error Rate
A	# Obstacle identification at normal orientation	98.2%	1.8%
	$ Obstacle identification at normal orientation	99.4%	0.6%
B	# Linear formation at normal orientation	94.6%	5.4%
	$ Linear formation at normal orientation	97.8%	2.2%
C	# Obstacle identification at angular orientation	78.8%	21.2%
	$ Obstacle identification at angular orientation	96.2%	3.8%
Transmission C to D	# Obstacle avoidance at angular orientation	79.4%	20.6%
	$ Obstacle avoidance at angular orientation	96.8%	3.2%
D	# flock formation	86.4%	13.6%
	$ flock formation	97.4%	2.6%

# Without servo motor integration, $ With servo motor integration.

## Data Availability

Data is available on https://www.youtube.com/watch?v=KmKvLk-DJOw (accessed on 11 October 2023).

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
