# Peer review of "A Versatile Approach to Polygonal Object Avoidance in Indoor Environments with Hardware Schemes Using an FPGA-Based Multi-Robot"

_sensors, 2023, doi:10.3390/s23239480_

Round 1

Reviewer 1 Report

Comments and Suggestions for Authors

The authors presented a novel design of hardware and algorithms for multi-robot autonomous navigation and obstacle avoidance in indoor environments. Overall, the reviewer feels it is an interesting study. However, there are several concerns that should be addressed before being accepted for publication as listed below:

1.       Line 123, Algorithm 1, it seems these criteria (“states”) are specifically designed to identify objects with one or multiple flat surfaces. Is this true? If not, please illustrate the capability of algorithms with more rich sources of objects (e.g., skeleton objects, concave objects)

2.       Line 132, the authors state “…SFT will estimate the depth of the object and SFM will establish the plane of the object (line 6)…” What is the rationale behind this design? Can S_FT be used to establish the plane, while S_FM be used to estimate the depth?

3.       Line 136, the authors state “In this environment, the objects are positioned with different angular orientations (line 9-13). The sensors estimate the object orientation with respect the present plane, and object identification performed in a similar way to that mentioned in lines 2-7.” It seems only the sofas and cupboards and triangular furniture are considered as obstacles for the algorithm validation. The reviewer is wondering if the algorithm can perform well for other objects such as pot plants, or chairs with a more irregular curved shape? If not, what is your thought in generalizing the algorithm on other objects?

4.       Line 152, what is the specification of the odometer sensors and how are used for localization? How does the flock or rendezvous group use the odometer sensors to localize all the robots? Please explain or provide a reference to these details.

5.       From Line 147-174, The reviewer found it is difficult to follow these texts. With reference to the text description, the authors should provide a more detailed schematic of Figure 2. For example, clearly label the left and right flock with the turning angle, and their trajectory path around the obstacles. In addition, with reference to these texts, the authors should supplement one additional figure which clearly shows the functionality of all sensors used with respect to a sample obstacle. Figure 1 is a good starting point but it is too simple.

6.       Line 200, the sentence seems grammatically incorrect. Please check.

7.       Figure 3, what does “ack” mean? Please explain.

8.       Section 2.2. The authors claim the scope of the paper is focus on static obstacles. Without carrying out additional experiments, can the authors provide a future vision on how the proposed method may be adapted to deal with dynamic obstacles?

9.       Line 242, the authors state “A human walks at a normal speed 242 of 1.44 m/sec, and the other robots’ speed is known information” Are these strict requirements or assumptions for the algorithm to perform well? Will one robot slowing down fail the algorithm?

10.   Section 3.2, the results section seems very short and simple. Can the authors supplement more findings such as the error analysis of localization, obstacle identification, and orientation estimation, with respect to different layouts of the obstacles and environments to form a more conclusive set of experiments? The results section in its current form does not well reflect how accurate and robust of the proposed system is.

Comments on the Quality of English Language

There exist minor grammar mistakes. The authors may improve the write-ups with some grammar-checking software

Author Response

Dear Reviewer, we thankful for your kind comments and suggestion. PFA of the responses for your kind comments. 

Reviewer 2 Report

Comments and Suggestions for Authors

The work shows an FPGA architecture for robot object avoidance tasks. There are some serious flaws in the paper.

- Why FPGA is needed?  Can a microcontroller do the same job? How about the performance and power consumption?

- There is no detailed explanation of the considered robots. A new section must be added describing in detail all the parts, sensors placements etc.

- Algorithms 1 and 2 represent FSM. Some state transition diagrams would be much more better.

- Section 2.1 looks like a wall of text. Please consider rewriting it.

- I did not get what the Authors mean by "pipeline". Are we talking about the registers used to increase the maximum clock frequency?

- Figure 3b is not so informative, it looks like a traditional ACK-based telecom protocol.

- What is PWDC sensor fusion? How is it  performed?

- "%" in y axis is missing in Fig. 9

- Which is the maximum clock frequency of the system? How many "evaluation" is the system able to perform in one second? Which is the "thsoughput"-

- A detailed discussion on power consumption is missing.

- The experimental results sections is missing any figure of merit. So, it is impossibile to compare to the state of the art, Table 2 is absolutely not enough.

Author Response

Dear Reviewer, we are thankful for your comments and suggestions. PFA of the responses as per your kind comments. 

Round 2

Reviewer 1 Report

Comments and Suggestions for Authors

I read the authors' response and the revised manuscript. The manuscript has been greatly improved. The reviewer has no further comments.

Reviewer 2 Report

Comments and Suggestions for Authors

I would like to thank the Authors for their responses. The paper may be considered for publication.